# Finding the Cracks: Improving LLMs Reasoning with Paraphrastic Probing and Consistency Verification

## Abstract

Large language models (LLMs) have demonstrated impressive performance across a variety of reasoning tasks in domains such as mathematics, coding, and planning, particularly when guided by chain-of-thought prompting to elicit intermediate reasoning steps. However, their problem-solving ability often declines on more complex tasks due to hallucinations and the accumulation of errors within these intermediate steps. Recent work has introduced the notion of *critical tokens*–tokens in the reasoning process that exert significant influence on subsequent steps. Prior empirical studies suggest that replacing critical tokens can refine reasoning trajectories and lead to correct answers. Nonetheless, reliably identifying and exploiting critical tokens to enhance LLM reasoning remains challenging. To address this, we propose the **P**araphrastic **P**robing and **C**onsistency **V**erification (**PPCV**) framework, which leverages critical tokens to improve reasoning performance. PPCV operates in two stages. In the first stage, we roll out an initial reasoning path from the original question and then concatenate paraphrased versions of the question with this reasoning path. Feeding these inputs into the LLM yields token-level logits, from which we identify critical tokens based on mismatches between the predicted top-1 token and the expected token in the reasoning path. A criterion is employed to confirm the final critical token. In the second stage, we substitute critical tokens with candidate alternatives and roll out new reasoning paths for both the original and paraphrased questions. The final answer is determined by checking the consistency of outputs across these parallel reasoning processes. We evaluate PPCV on mainstream LLMs, including Llama-3.1-8B-Instruct, Mistral-7B-Instruct-v0.2 and Qwen3-32B, across multiple benchmarks covering mathematics and logical reasoning. Extensive experiments demonstrate that PPCV substantially enhances the reasoning performance of LLMs compared to baseline methods.

## 1 Introduction

The emergence of large language models (LLMs) (Brown et al., 2020; Grattafiori et al., 2024; Achiam et al., 2023; Yang et al., 2025a) has astonished the AI community with their remarkable capabilities across a wide range of reasoning tasks, including mathematical problem solving, programming, and planning. By generating intermediate reasoning steps through techniques such as chain-of-thought prompting (Wei et al., 2022; Kojima et al., 2022; Zhang et al., 2022; Sprague et al., 2025), LLMs can emulate human-like reasoning processes and achieve strong performance on diverse reasoning benchmarks.

Despite their success, the problem-solving ability of LLMs often declines on complex reasoning tasks due to hallucinations and the accumulation of errors in intermediate steps (Lightman et al., 2023; Ling et al., 2023;

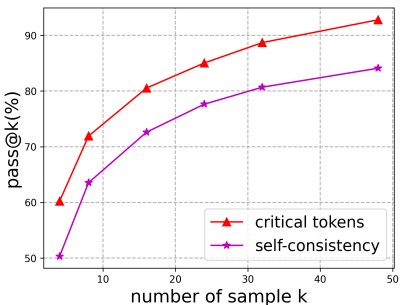

Figure 1: Comparison of the effects of critical tokens and Self-Consistency on the reasoning performance of LLMs, evaluated on samples from the MATH training data.

Bubeck et al., 2023). To mitigate this issue, prior works (Wang et al., 2024; Yuan et al., 2025; Chen et al., 2024a; Chen & Li, 2024) have proposed the inference-time optimization paradigm, which evaluates and refines reasoning trajectories through self-correction by the LLM itself or with feedback from external verifiers such as process reward models (PRMs) (Uesato et al., 2022). However, existing studies (Yang et al., 2025b; Tyen et al., 2024; Stechly et al., 2025) suggest that LLMs struggle to reliably identify errors in their reasoning and often fail to correct previous outputs without external guidance. Moreover, obtaining high-quality, stepwise process supervision for training external verifiers is challenging and limits the practicality of these approaches (Feng et al., 2025).

Recent work has introduced the concept of *critical tokens* (Lin et al., 2025), which play pivotal roles in intermediate reasoning steps and exert strong influence on subsequent reasoning and final outcomes. Prior studies suggest that critical tokens often diverge from human-annotated error tokens. Moreover, as illustrated in Figure 2, replacing critical tokens in an incorrect reasoning trajectory with suitable candidate tokens can correct subsequent steps and lead to the right answer in new rollouts. To quantitatively assess the effectiveness of critical tokens, we conduct an empirical study using LLMs such as Llama-3.1-8B-Instruct (Grattafiori et al., 2024) on reasoning tasks. Specifically, we randomly sample 100 instances with incorrect reasoning steps from the GSM8K (Cobbe et al., 2021) and Math500 (Hendrycks et al., 2021) training data. Following the procedure in prior work, we lo-

**Case study on critical tokens**

**Question:** There were 10000 spectators at a basketball match. 7000 of them were men. Of the remaining spectators, there were 5 times as many children as women. How many children were there? (ground truth answer: 2500)
**Response:**
To find the number of children, we need to first find the number of **women** and then multiply it by 5.
Step 1: Find the number of women. Total spectators = 10000. Men = 7000. Women = Total spectators – Men. Women = 10000 – 7000. Women = 3000.
Step 2: Find the number of children. There are 5 times as many children as women. Children = 5 * Women. Children = 5 * 3000. Children = 15000.
So, the number of children is 15000. ✖

**Response(replacement of critical tokens):**
To find the number of children, we need to first find the number of **remaining** spectators (excluding men) and then calculate the number of women and children.
Step 1: Find the number of remaining spectators (excluding men). Total spectators = 10000. Men = 7000. Remaining spectators = Total spectators – Men = 10000 - 7000= 3000
Step 2: Let's assume the number of women is 'x'. Since there are 5 times as many children as women, the number of children will be 5x.
Step 3: The total number of remaining spectators is the sum of women and children. We can write this as an equation: x + 5x = 3000. Combine like terms:\n6x = 3000. Divide both sides by 6: x = 3000 / 6. x = 500.
Step 4: Now that we have the number of women, we can find the number of children. Number of children = 5x= 5 * 500= 2500.
Therefore, the number of children is 2500. ✔

Figure 2: An example demonstrating how substitution of a critical token (red) with a candidate token (purple) modifies the reasoning path and produces the correct answer.

cate critical tokens through exhaustive search. We then truncate the reasoning path at the critical token, substitute it with alternative tokens, and roll out new reasoning paths. For example, as shown in Figure 2, the token "woman" is replaced with "remaining". We evaluate performance using Pass@k and compare against Self-Consistency (Wang et al., 2023), which also samples multiple reasoning paths. As shown in Figure 1, critical token replacement provides a clear advantage in correcting errors compared to plain sampling. Nonetheless, reliably identifying and leveraging critical tokens for reasoning remains a nontrivial challenge. Additional results can be found in Appendix A.1.

Recent studies (Zhou et al., 2024; Chen et al., 2024b), on *surface form*—the way questions, assumptions, and constraints are phrased—have revealed its subtle influence on the trajectory of intermediate reasoning steps. This motivates us to explore the role of paraphrasing in the extraction and utilization of critical tokens for reasoning tasks. To this end, we propose the **P**araphrastic **P**robing and **C**onsistency **V**erification (**PPCV**) framework , a two-stage approach designed to leverage critical tokens to enhance the reasoning ability of LLMs. In the first stage, we probe critical tokens using paraphrased questions. Specifically, we first roll out the initial reasoning path from the original question, then concatenate paraphrased questions with this reasoning path. The resulting synthetic inputs are fed into the LLM to obtain token-level logits for each position in the reasoning path. Positions where the predicted top-1 token diverges from the expected token are flagged as potential pivotal points, and an empirical criterion is applied to determine the final critical token. In contrast to prior work (Lin et al., 2025), which depends on external models for identifying critical tokens with ambiguous criteria, our method introduces a self-contained mechanism that pinpoints critical tokens.

In the second stage, we leverage the extracted critical tokens to refine the initial reasoning path. Specifically, we select the top-K tokens (include critical token itself) at the critical token position and roll out new reasoning paths for both the original and paraphrased questions. Based on the empirical observation that trajectories leading to correct answers are robust to paraphrastic perturbations, we propose a paraphrase consistency mechanism. In contrast to Self-Consistency (Wang et al., 2023), which relies on majority voting across sampled trajectories, our method selects the final answer by comparing outputs from paraphrased and original questions and choosing the one with the most consistent matches. To address potential ties across multiple answers, we further in-

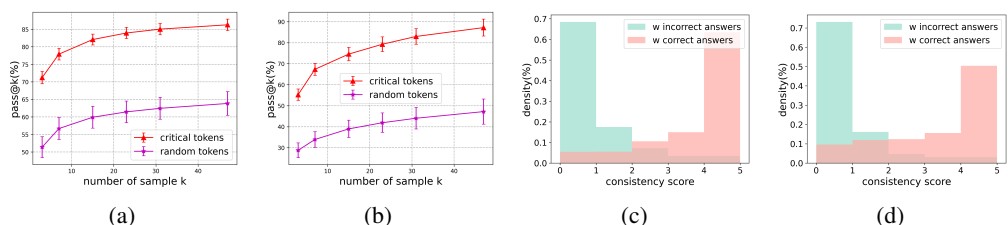

(a)          (b)          (c)          (d)

Figure 3: Comparison of the impact of critical tokens versus random tokens on LLM reasoning performance for GSM8K (a) and MATH (b). Comparison of the density distributions of consistency scores for rollouts with correct and incorrect answers on GSM8K (c) and MATH (d).

troduce similarity-weighted paraphrase consistency, which incorporates similarity scores between paraphrased and original questions when computing consistency.

Compared with self-correction (Wu et al., 2024; Miao et al., 2024) and PRM-based methods (Wang et al., 2024; Yuan et al., 2025), our framework exploits critical tokens to refine reasoning trajectories without requiring step-level error detection by the LLM itself or auxiliary models. We evaluate our method on mainstream LLMs across mathematical, logical, and commonsense reasoning benchmarks, demonstrating consistent improvements in reasoning performance. The contributions of the paper is summarized as follows:

- We propose a novel two-stage framework, **P**araphrastic **P**robing and **C**onsistency **V**erification (**PPCV**) designed to extract and leverage critical tokens to enhance the reasoning performance of LLMs.

- We show that critical tokens can more effectively correct erroneous reasoning trajectories than traditional sampling methods like Self-Consistency. Furthermore, our approach successfully extracts these tokens through paraphrastic probing, achieving improved final results via paraphrase consistency.

- We evaluate our method on mainstream LLMs across various reasoning tasks, including math and logic reasoning. Experimental results show significant performance improvements over baseline methods.

## 2 RELATED WORK

**Inference-Time Optimization for LLM reasoning.** With the advent of chain-of-thought (CoT) prompting, LLMs have demonstrated strong reasoning capabilities by producing intermediate steps during inference. This success has motivated a growing body of work (Wu et al., 2025; Snell et al., 2024) on augmenting reasoning trajectories at test time to further improve performance. Existing approaches can be broadly categorized into search-based methods (Bi et al., 2025; Yao et al., 2023; Hao et al., 2023; Xie et al., 2023; Besta et al., 2024), such as Tree-of-Thoughts (Yao et al., 2023), and sampling-based methods (Wang et al., 2023; Xu et al., 2025; Wan et al., 2025; Ma et al., 2025), such as Self-Consistency (Wang et al., 2023). However, due to intrinsic hallucinations (Bubeck et al., 2023), LLMs often generate erroneous intermediate steps, which can ultimately lead to incorrect answers, especially on complex problems. This limitation highlights the need for inference-time optimization of reasoning processes.

To address this issue, one line of research (Yin et al., 2024; Chen et al., 2024a; Ling et al., 2023; Wu et al., 2024; Miao et al., 2024; Madaan et al., 2023) designs instructional prompts that guide LLMs to detect and refine their own mistakes. Despite its appeal, prior work has shown that the effectiveness of self-correction is limited in practice. Another line of work (Wang et al., 2024; Yuan et al., 2025; He et al., 2024; Havrilla et al., 2024) introduces external verifiers, such as process reward models (PRMs) (Snell et al., 2024), to identify and filter out error-prone reasoning steps. These methods typically require high-quality training data for the verifier, with data scarcity often mitigated through strategies such as Monte Carlo Tree Search (MCTS) Guan et al. (2025); Qi et al. (2025); Li (2025); Zhang et al. (2024). In addition, a recent line of decoding-based approaches (Xu et al., 2025; Ma et al., 2025) seeks to improve reasoning by dynamically adjusting the next-token prediction based on future trajectory probing. In contrast, our method refines reasoning by directly

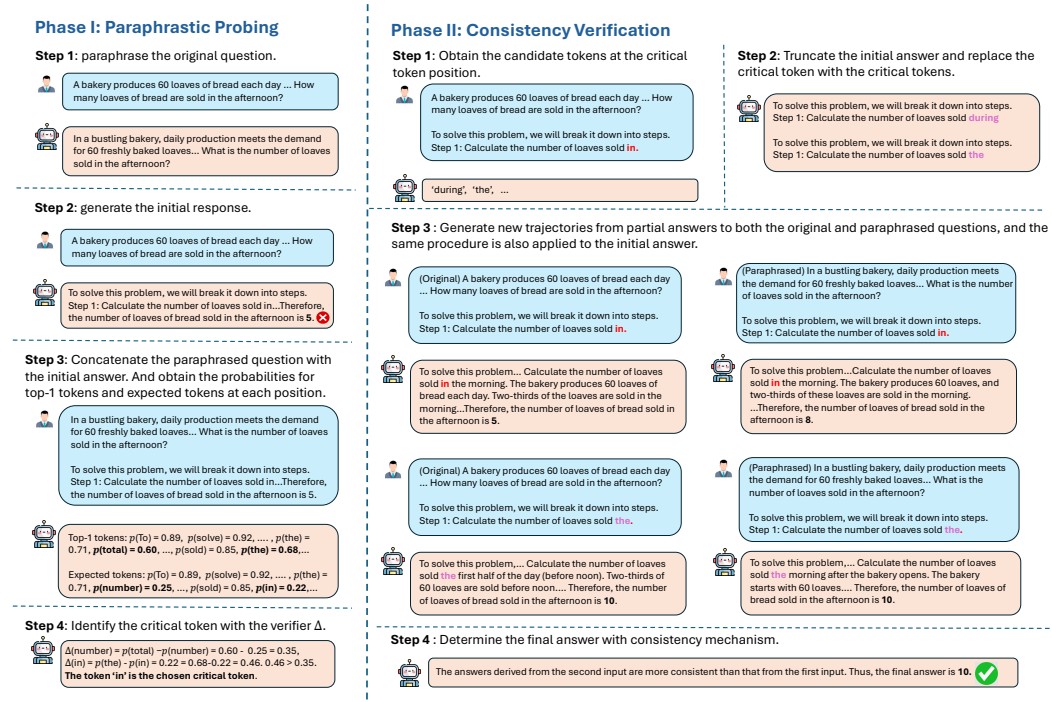

Figure 4: Illustration of the proposed paraphrastic probing and consistency verification (PPCV) framework. The framework comprises two stages: (i) probing critical tokens through paraphrased forms, and (ii) rolling out new reasoning steps with alternative tokens and selecting the final answer using the paraphrase consistency verification mechanism.

leveraging critical tokens, without relying on stepwise verification or external verifiers. This design underscores both the utility and universality of our approach.

**Paraphrasing for LLMs.** A growing number of work (Zhou et al., 2024; Chen et al., 2024b; Gao et al., 2024) has examined the impact of a problem's surface form on the reasoning ability of LLMs. Findings (Zhou et al., 2024; Chen et al., 2024b; Huang et al., 2025) suggest that even subtle modifications in phrasing can substantially affect both the reasoning process and the final outcome. Building on this observation, several methods (Yadav et al., 2024; Chen et al., 2024b) leverage beneficial paraphrasing to enhance LLM performance in tasks such as reasoning and intent classification. In addition, paraphrasing has been employed to assess model uncertainty (Gao et al., 2024; Tanneru et al., 2024), thereby enhancing the reliability and trustworthiness of LLM applications. In our work, we utilize paraphrasing as a principled tool to extract critical tokens and to aggregate answers.

## 3 METHODOLOGY

In this section, we present the two components of our framework in detail: paraphrastic probing and consistency verification. An detailed illustration of our framework is shown in Figure 4. We then discuss the proposed method and provide the complete algorithm.

### 3.1 PARAPHRASTIC PROBING

Previous findings (Zhou et al., 2024; Chen et al., 2024b; Huang et al., 2025) on the impact of a problem's surface form suggest that the quality of intermediate reasoning steps is influenced not only by the underlying mathematical relationships and logic, but also by how the problem is expressed. Notably, LLMs are sometimes able to solve a paraphrased version of a problem that they fail to solve in its original form, highlighting the potential of paraphrasing to uncover pivotal tokens that are critical for successful reasoning. Motivated by this observation, we introduce paraphrastic probing to efficiently identify the critical token. Given the original question $q_0$, we first prompt the LLM to generate multiple paraphrased forms, denoted as $q_1, q_2, ..., q_N$, where $N$ is the number of the

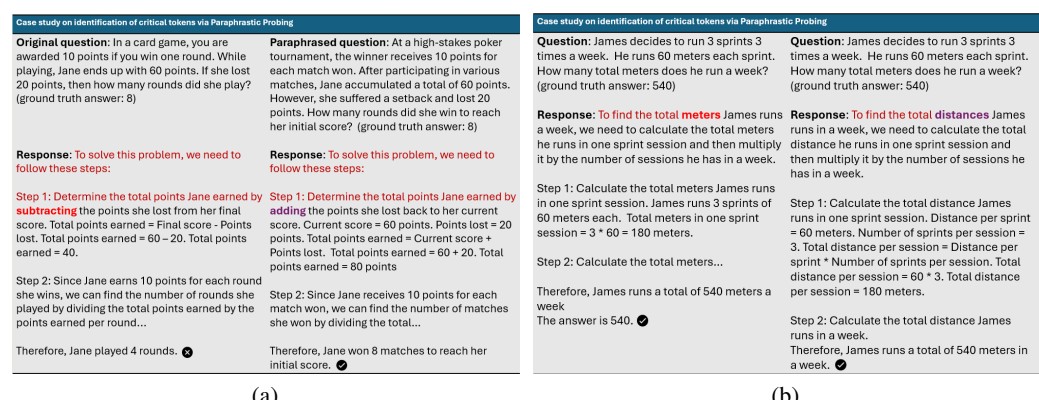

Figure 5: (a) Illustration of paraphrastic probing for critical token identification. (b) Effect of the critical token on an initial reasoning path that yields the correct answer. Critical tokens are highlighted in bold red, and alternative tokens in bold purple.

paraphrased questions. We adopt Automatic Prompt Engineering (APE) (Zhou et al., 2022) to derive paraphrasing instructions that strictly preserve all numerical values, mathematical relationships, and core logical structures of the problem, while maximizing linguistic and contextual diversity. Additional details can be found in Appendix A.2. We then obtain the initial reasoning path $r_0^{q_0}$ for the original question using greedy decoding. This reasoning path is subsequently concatenated with each paraphrased question, and the resulting synthetic inputs are fed into the LLM to compute the token probability distribution at each position in $r_0^{q_0}$. Specifically, the token probability distribution at $i$th position conditioned on the paraphrased question $q_n$ is expressed as

$$P_i^{q_n} = \text{LLM}(\tilde{a}_i|\mathcal{I}, q_n, r_{0,<i}^{q_0}), \tag{1}$$

where $\mathcal{I}$ denotes the instruction prefix and $\tilde{a}_i$ represents the sampled token at $i$th position. The token $\tilde{a}_i$ is regarded as a candidate critical token if predicted top-1 token does not match the expected token at the same position in $r_0^{q_0}$, i.e.,

$$\arg\max P_i^{q_n} \neq a_i, \tag{2}$$

where $a_i$ denotes the expected token at the $i$th position in $r_0^{q_0}$.

To validate the effectiveness of our method and demonstrate the pivotal role of the critical tokens extracted through paraphrastic probing, we conduct a case study illustrated in Figure 5(a). In this example, the token "subtracting" is identified as a critical token. By substituting it with an alternative (i.e., "adding"), the new rollout corrects the errors in the original reasoning steps and yields the correct answer, underscoring the utility of our method in identifying critical tokens. In addition, we conduct a quantitative analysis to investigate the impact of critical tokens extracted through paraphrastic probing, comparing them against randomly selected tokens. Specifically, we sample 100 instances with incorrect reasoning trajectories from the GSM8K (Cobbe et al., 2021) and MATH500 (Hendrycks et al., 2021) training sets. Following the paraphrastic probing pipeline, we identify candidate critical tokens in the initial reasoning steps, substitute them with alternative tokens, and roll out multiple new reasoning paths. We randomly sample 40% of the candidate critical tokens for evaluation in each run and repeat the experiments 10 times independently. For comparison, we apply the same procedure to randomly chosen tokens. We evaluate performance using pass@k on Llama-3.1-8B-Instruct (Grattafiori et al., 2024), with results presented in Figure 3(a) and Figure 3(b). The findings demonstrate that substituting critical tokens significantly improves reasoning performance compared to random tokens, thereby further validating the pivotal role extracted critical tokens as well as the effectiveness of our method.

We introduce a heuristic verifier to select the final critical token from multiple candidates. For a candidate token $a_i$ and paraphrased question $q_n$, the verification score is defined as

$$\Delta_{q_n}(a_i) = \max P_i^{q_n} - P_i^{q_n}(\tilde{a}_i = a_i). \tag{3}$$

where $P_i^{q_n}$ denotes the predictive distribution at position $i$ on question $q_n$. Intuitively, $\Delta$ measures how much the predicted top-1 token deviates from the expected token, indicating the token's potential impact on the reasoning trajectory. For each extracted token $a_i$, we take the maximum score

---

**Algorithm 1** PPCV Framework

---

**Input:** LLM; original question $q_0$; number of paraphrased questions $N$; number of sampled alternative tokens $K$; temperature-scaling coefficient $\lambda$.

**Output:** Final answer ans$_f$.

    (**Paraphrasing**) Generate paraphrased variants $q_1, q_2, \ldots, q_N$ from $q_0$ using the LLM.

    (**Initial reasoning**) Obtain the initial reasoning path $r_0^{q_0}$ for $q_0$.

    (**Token influence analysis**) For each paraphrased question $q_n$, compute the token distribution $P^{q_n}$ by concatenating $q_n$ with $r_0^{q_0}$ (Eq. 1).

    (**Candidate selection**) Identify candidate critical token positions based on the mismatch between the top-1 predicted tokens and the expected tokens in $q_0$ (Eq. 2).

    (**Critical token verification**) Select the final critical token $a_c$ using the criteria (Eqs. 3–5).

    (**Alternative token sampling**) Sample $K$ alternative tokens $a_c^0, a_c^1, \ldots, a_c^{K-1}$ (including $a_c^0 = a_c$) using top-$K$ sampling on $q_0$.

    (**Truncated rollouts**) Truncate the initial reasoning path $r_0^{q_0}$ at position $a_c$ and form synthetic inputs $\tilde{r}_c^0, \tilde{r}_c^1, \ldots, \tilde{r}_c^{K-1}$ by appending each alternative token.

    **for** each synthetic input $\tilde{r}_c^k, k = 0, \ldots, K - 1$ **do**

        Generate rollouts $r_k^{q_0}, r_k^{q_1}, \ldots, r_k^{q_N}$ for the original and paraphrased questions using the LLM.

    **end for**

    (**Consistency verification**) Compute the final answer ans$_f$ using (similarity-weighted) paraphrase consistency (Eq. 6).

---

across paraphrases,

$$\Delta(a_i) = \max_{q_n} \Delta_{q_n}(a_i), \tag{4}$$

and select the final critical token as

$$a_c = \arg\max_i \Delta(a_i). \tag{5}$$

## 3.2 CONSISTENCY VERIFICATION

After identifying the final critical token $a_c$, we aim to refine the original reasoning path with alternative tokens and achieve final answer with paraphrase consistency mechanism. Specifically, we generate a set of alternative tokens $a_c^0, a_c^1, a_c^2, \ldots, a_c^{K-1}$ using the LLM conditioned on original question $q_0$, where $a_c^0$ is the original token in $r_0^{q_0}$ and the remaining tokens are sampled via top-K sampling. The initial reasoning path is truncated at the position of critical token, and each alternative token is concatenated to form synthetic inputs $\tilde{r}_c^0, \tilde{r}_c^1, \tilde{r}_c^2, \ldots, \tilde{r}_c^{K-1}$. We then roll out new reasoning trajectories for each synthetic input with respect to both the original and paraphrased questions using greedy decoding, denoted as $r_k^{q_0}, r_k^{q_1}, \ldots, r_k^{q_N}$ for $k = 0, 1, 2, \ldots, K - 1$. Next, for the rollout with the $k$th alternative token, we compare the answers obtained from the paraphrased forms with that from the original form and compute a consistency score $c_k = \sum_{n-1}^N \mathbb{I}(\Phi(r_k^{q_0}) = \Phi(r_k^{q_n}))$, where $\Phi(\cdot)$ and $\mathbb{I}(\cdot)$ denotes the function that extracts the final answer from a reasoning trajectory and the indicator function, respectively. The answer associated with the highest consistency score is then selected as the final prediction

$$\text{ans}_f = \Phi(r_k^{q_0}), \text{where } k = \arg\max_k c_k. \tag{6}$$

To justify our paraphrase consistency mechanism, we investigate the impact of paraphrased forms on LLM reasoning. We sample instances from the GSM8K (Cobbe et al., 2021) and MATH training sets (Hendrycks et al., 2021) and follow our pipeline to extract critical tokens. From each truncated reasoning trajectory, we roll out multiple reasoning paths by concatenating alternative tokens. For each original question, we generate five paraphrased variants and compute the consistency score for resulting rollouts. The evaluation is conducted on Llama-3.1-8B-Instruct (Grattafiori et al., 2024). We then analyze the distribution of consistency scores for rollouts that yield correct versus incorrect answers. As shown in Figure 3(c) and Figure 3(d), more than 90% of rollouts with correct answers achieve a consistency score of at least 1, whereas this proportion drops to around 30% for rollouts with incorrect answers. This sharp contrast indicates that correct rollouts are more robust across paraphrased variants, motivating the design of our paraphrase consistency mechanism to exploit this property for improved final predictions.

Table 1: Comparison of our method with baseline approaches on Llama-3.1-8B-Instruct and Mistral-7B-Instruct-v0.2.

| Model | Method | GSM8K | GSM-Hard | Math500 | SVAMP | ARC |
|---|---|---|---|---|---|---|
| Llama-3.1-8B-Instruct | Chain-of-Thought | 77.40 | 28.00 | 31.00 | 83.00 | 58.91 |
| | Self-Consistency | 80.60 | 31.80 | 37.80 | 85.10 | 60.75 |
| | Tree-of-Thought | 75.74 | 33.28 | 31.60 | 81.20 | 80.72 |
| | Guided Decoding | 75.51 | 32.45 | 31.20 | 81.70 | 81.74 |
| | Predictive Decoding | 81.43 | 40.26 | 34.00 | 85.90 | 84.56 |
| | Phi-Decoding | 86.58 | 39.88 | 38.20 | 84.50 | 85.41 |
| | PPCV (Ours) | **88.24** | **49.73** | **50.00** | **89.60** | **88.31** |
| Mistral-7B-Instruct-v0.2 | Chain-of-Thought | 46.45 | 26.91 | 12.20 | 62.40 | 41.42 |
| | Self-Consistency | 50.38 | 28.65 | 14.20 | 66.70 | 44.54 |
| | Tree-of-Thought | 50.49 | 25.78 | 11.40 | 60.60 | 41.04 |
| | Guided Decoding | 50.79 | 27.07 | 14.00 | 62.90 | 39.51 |
| | Predictive Decoding | 55.67 | 27.07 | 14.40 | 62.10 | 47.87 |
| | Phi-Decoding | **56.60** | 28.43 | 13.40 | 63.20 | 60.24 |
| | PPCV (Ours) | 56.58 | **31.08** | **14.60** | **69.30** | **69.88** |

Table 2: Comparison of our method with baseline approaches on Qwen3-32B (non-thinking mode).

| Model | Method | AIME2024 | AIME2025 | BRUMO2025 | HMMT2025 |
|---|---|---|---|---|---|
| Qwen3-32B | Chain-of-Thought | 30.00 | 23.67 | 30.00 | 9.67 |
| | Guided Decoding | 26.67 | 22.67 | 28.67 | 7.33 |
| | Predictive Decoding | 32.67 | 24.00 | 33.33 | 10.33 |
| | Phi-Decoding | 33.60 | 24.33 | 36.67 | 10.67 |
| | PPCV (Ours) | **40.00** | **26.00** | **43.33** | **13.33** |

To address potential collisions when multiple answers obtain the same maximum consistency score, we introduce similarity-weighted consistency verification. Inspired by weighted majority voting (Dogan & Birant, 2019), this approach adjusts the influence of each paraphrased form on the consistency score according to its similarity to the original form. Intuitively, paraphrased forms with lower similarity should exert greater weight, as they provide stronger evidence of robustness, whereas those closely resembling the original form contribute less. Concretely, we first extract embeddings for both the original and paraphrased questions and compute their similarity scores as $s_n = \text{sim}(q_0, q_n)$, where $\text{sim}(\cdot)$ denotes a similarity measure such as cosine similarity. We then derive weights via a softmax function $w_n = \text{softmax}(s_n) = \frac{\exp(-\lambda s_n)}{\sum_n \exp(-\lambda n)}$, where $\lambda$ is the temperature scaling coefficient. Finally, the similarity-weighted consistency score is defined as $\tilde{c}_k = \sum_{n-1}^{N} w_n \mathbb{I}(\Phi(r_k^{q_0}) = \Phi(r_k^{q_n}))$. This ensures agreement with more diverse paraphrases contributes more strongly to the final decision.

## 3.3 DISCUSSION

We have shown that replacing critical tokens can correct intermediate reasoning paths and lead to the correct answer. In this section, we examine how our method influences reasoning paths that are already correct. First, we conduct a case study on an instance with a correct answer, where we follow our pipeline to identify the critical token and roll out new reasoning paths using alternative tokens. As illustrated in Figure 5(b), the new rollouts also yield the correct answer. Second, our pipeline incorporates both the initial reasoning path $r_0^{q_0}$ and its paraphrased variants $r_0^{q_n}$ for evaluation. The robustness of correct rollouts across paraphrased forms ensures high consistency scores, allowing them to stand out as the final answer. These findings suggest that our pipeline preserves the performance of LLMs on problems that can already be solved correctly by CoT (Wei et al., 2022). Finally, the complete algorithm of our proposed PPCV framework is illustrated in Algo. 1.

Our technical contributions differ from prior works in three distinct ways. First, prior works (Zhou et al., 2024; Chen et al., 2024b; Yadav et al., 2024) typically use paraphrasing merely to expand the solution space. In contrast, we introduce Paraphrastic Probing, a mechanism that uses paraphrasing to test the model's internal confidence. By analyzing the discrepancy in token-level logits of the initial trajectory between the original and paraphrased questions, we can rigorously pinpoint the critical tokens that may lead to errors in the following steps.This transforms paraphrasing from a generation tool into a precise, token-level diagnostic tool. Second, prior works (Zhou et al., 2024;

Table 3: Comparison of model performance when using critical tokens versus random tokens.

| Method | GSM8K | GSM-Hard | Math500 | SVAMP | ARC |
|---|---|---|---|---|---|
| Chain-of-Thought | 77.40 | 28.00 | 31.00 | 83.00 | 58.91 |
| random tokens | 82.08 | 40.29 | 42.12 | 84.77 | 75.68 |
| critical tokens (Ours) | **88.24** | **49.73** | **50.00** | **89.60** | **88.31** |

Table 4: Comparison of our proposed paraphrase consistency against the majority voting.

| Method | GSM8K | GSM-Hard | Math500 | SVAMP | ARC |
|---|---|---|---|---|---|
| Chain-of-Thought | 77.40 | 28.00 | 31.00 | 83.00 | 58.91 |
| majority voting | 87.20 | 47.36 | 48.19 | 88.80 | 86.16 |
| paraphrase consistency (Ours) | **88.24** | **49.73** | **50.00** | **89.60** | **88.31** |

Chen et al., 2024b) typically rely on simple majority voting across multiple solutions. Our Paraphrase Consistency mechanism is technically distinct. It validates answers based on their robustness across semantic variations of the problem constraint. We further introduce a similarity-weighted consistency metric that weighs answers based on the linguistic diversity of the paraphrase, offering a more nuanced selection criterion than simple frequency counts. At last, a major technical limitation in current reasoning research is the reliance on external models or human-annotated error steps. Our method contributes a fully self-contained pipeline that identifies and corrects errors using the model's own sensitivity to surface-form perturbations. We believe our proposed framework represents a significant methodological advancement, offering a way to "find the cracks" in reasoning without the heavy computational or data overhead of training separate verifiers.

Although we select the top candidate for the primary experiments to maintain computational efficiency, the framework itself naturally extends to the multi–critical-token setting. For multiple critical tokens, we can generate alternative tokens for each identified position and apply paraphrase consistency across the new rollouts. This allows the model to refine multiple segments of its intermediate reasoning steps rather than only one.

## 4 EXPERIMENTS

In this section, we first describe the experimental setup, followed by the main results of our proposed method compared to the baselines. We then analyze the contribution of each stage individually. Finally, we examine the actual running time of our method and compare it against the baselines.

### 4.1 SETUP

**Datasets**. To comprehensively assess our method, we evaluate it on seven benchmarks. Six focus on mathematical reasoning, including GSM8K (Cobbe et al., 2021), GSM-Hard (Gao et al., 2023), SVAMP (Patel et al., 2021), Math500 (Hendrycks et al., 2021), and the more challenging competition-level datasets AIME2024, AIME2025, BRUMO2025, and HMMT2025 (Balunović et al., 2025). In addition, we use ARC-Challenge (Clark et al., 2018) to evaluate knowledge reasoning ability of large language models.

**Baselines**. In our experiments, we use Chain-of-Thought (CoT) (Wei et al., 2022), Tree-of-Thought (ToT) (Yao et al., 2023), Guided Decoding (Xie et al., 2023), Predictive Decoding (Ma et al., 2025), and Phi-Decoding (Xu et al., 2025) as baselines.

**Metric**. Following prior work, we adopt pass@1 accuracy as the primary evaluation metric.

**Implementation Details**. In our experiments, we adopt Llama-3.1-8B-Instruct (Grattafiori et al., 2024), Mistral-7B-Instruct-v0.2 (Jiang et al., 2023), Qwen-3-32B (Yang et al., 2025a) and DeepSeek-R1-Distill-Llama-70B as the target models. we employ the non-thinking mode for Qwen-3-32B. Throughout our method, we employ the same model for generating paraphrased problems, identifying critical tokens, and producing new rollouts. In the first stage, we generate three paraphrased variants of each problem, while for the other datasets we generate five variants. In the second stage, we select the top 10 tokens for new rollouts, with the temperature scaling coefficient $\lambda$ set to 2. We also adopt a zero-shot CoT prompt to elicit the new rollouts. For the baselines,

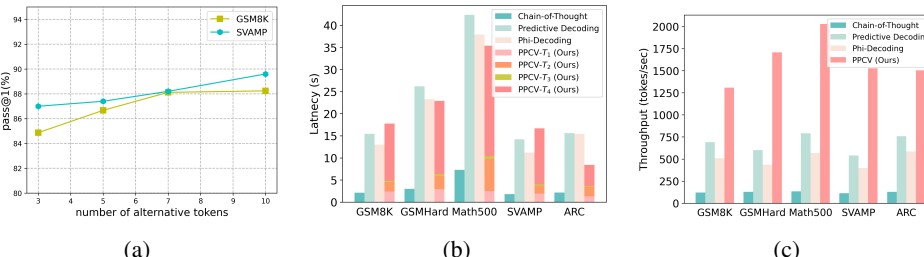

(a)                                        (b)                                        (c)

Figure 6: (a) The impact of number of sampled alternative tokens on the performance. (b) Latency comparison between the baselines and our method, measured as the average inference time per question (in seconds). $T_1, T_2, T_3, T_4$ denote time for paraphrased question generation, initial answer generation, forward pass and new rollouts from truncated trajectories. (c) Throughput comparison between the baselines and our method, measured in output tokens per second.

we strictly follow their original settings, including temperature values, sampling strategies, and the number of few-shot examples. All experiments are conducted on NVIDIA A100 GPUs.

## 4.2 MAIN RESULTS

The main results are summarized in Table 1 and Table 2. The results indicate that Self-Consistency effectively improves the reasoning performance of LLMs compared to vanilla Chain-of-Thought prompting. For example, Llama-3.1-8B-Instruct (Grattafiori et al., 2024) achieves about 3% higher accuracy with Self-Consistency than with CoT. These findings suggest that augmenting reasoning during inference through sampling is an effective way to refine reasoning trajectories. Recent decoding-based methods, such as Predictive Decoding (Ma et al., 2025) and Phi-Decoding (Xu et al., 2025), also achieve strong results.Unlike prior works that rely on carefully designed prompts to self-correct errors in intermediate steps, these two methods modify the current step by probing future steps with pre-defined reward signals. Furthermore, our experimental results demonstrate that the proposed method consistently outperforms the baselines across most tasks, spanning both mathematical and knowledge reasoning, thereby highlighting its generalization ability across different reasoning settings. Notably, our method even surpasses the latest approaches such as Predictive Decoding (Ma et al., 2025) and Phi-Decoding (Xu et al., 2025). In particular, it achieves approximately 50.00% accuracy on the Math500 dataset (Hendrycks et al., 2021), exceeding these baselines considerably. The results on competition-level datasets further demonstrate the effectiveness of our method in enhancing the reasoning ability of LLMs. These results indicate that our method can effectively extract critical tokens that play a pivotal role in the final outcome and correct the reasoning trajectory by leveraging alternative tokens. Additional results can be found in Appendix A.3.

## 4.3 ABLATION STUDY

In this section, we analyze the contribution of each stage individually. Additionally, we investigate how the number of sampled alternative tokens affects the overall performance. All the evaluations are conducted on Llama-3.1-8B-Instruct (Grattafiori et al., 2024). Additional ablation study can be found in Appendix A.4.

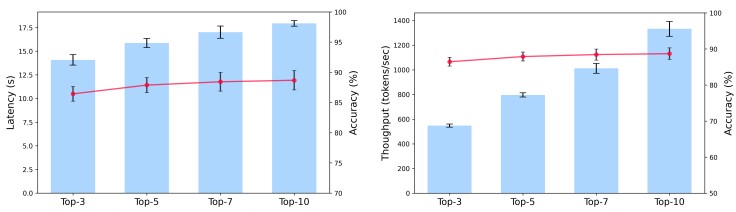

Figure 7: Trade-off between performance and efficiency for our method under different numbers of top-k tokens (k = 3, 5, 7, 10), measured by latency (left) and throughput (right).

**Effectiveness of extracted critical tokens**. To demonstrate the effectiveness of our extracted critical tokens, we conduct an evaluation in which the critical tokens are replaced with random tokens in the first stage, while keeping the second stage unchanged. This evaluation is performed across multiple benchmark datasets, with pass@1 as the metric. The results, shown in Table 3, reveal a substantial

decline in performance. These findings highlight the pivotal role of critical tokens and indicate that our method can effectively identify and extract them.

**Effectiveness of paraphrase consistency**. We also evaluate the effectiveness of our proposed paraphrase consistency and compare it with traditional majority voting. While keeping the first stage unchanged, instead of using paraphrased forms to generate new reasoning steps, we simply sample multiple new steps from alternative tokens conditioned on the original question and use majority voting to determine the final answer. The results, shown in Table 4, reveal a noticeable decline in performance, highlighting the importance of paraphrased forms in improving the intermediate reasoning steps.

**Impact of number of sampled alternative tokens**.We investigate the influence of the number of sampled alternative tokens in the second stage by selecting values of 3, 5, 7, and 10. The results, shown in Figure 6(a), demonstrate that performance improves as the number of alternative tokens increases. This suggests that exploring more reasoning steps with additional alternative tokens during inference can be beneficial for reasoning tasks.

## 5 COMPUTATIONAL COST ANALYSIS

In this section we examine the composition of the latency in our method. The latency arises from four components: Paraphrased question generation ($T_1$); initial answer generation ($T_2$), equivalent to vanilla CoT; a forward pass for identifying critical tokens ($T_3$), which does not generate new tokens and is computationally lightweight; rollouts of truncated trajectories using alternative tokens under both the original and paraphrased questions ($T_4$), which constitutes the main source of overhead.

We evaluate all components on Llama-3.1-8B-Instruct using vLLM on NVIDIA A100 GPUs, with a maximum output length of 4096 tokens for each question. For our method, we use 5 paraphrased questions on math datasets and 3 on ARC, and select the top-10 candidate tokens as alternatives. The updated average latency results are reported in Figure 6(b). As expected, $T_1$ scales with the number of paraphrases, $T_3$ remains minimal, and $T_4$ dominates the total cost. Specifically, $T_4$ depends on the number of top-k alternative tokens, the number of paraphrased questions and the position of the critical token in the trajectory. Since the new rollouts from truncated trajectories for different alternative tokens and paraphrased questions are independent, $T_4$ can take advantage of vLLM's parallelism. These rollouts can therefore be processed concurrently, improving overall efficiency. This is reflected in the higher throughput (tokens/sec) shown in Figure 6(c). And results of our method in latency comparable to baseline methods, even on challenging benchmarks such as MATH500 and GSM-Hard where the critical token tends to occur in later reasoning steps. On benchmarks such as GSM8K and SVAMP, both our method and baselines like Predictive Decoding exhibit higher latency compared to vanilla Chain-of-Thought.

We also conduct a trade-off analysis between performance and efficiency by varying the number of alternative tokens considered at the critical position. Specifically, we test top-3, top-5, top-7, and top-10 alternatives using GSM8K, sampling 200 random questions and repeating the experiment five times to compute confidence intervals. The results are presented in Figure 7. We observe a clear trade-off. Reducing the number of alternative tokens lowers both latency and throughput, while causing a slight reduction in accuracy. This provides a practical mechanism for adjusting performance-efficiency trade-offs in real deployments. Depending on resource availability and target accuracy, practitioners can choose the appropriate number of alternative tokens.

## 6 CONCLUSION

In this study, we further investigate the pivotal role of critical tokens in shaping the reasoning trajectory, as well as the beneficial impact of paraphrase forms on reasoning. To leverage these two factors, we propose the Paraphrastic Probing and Consistency Verification (PPCV) framework. Our framework consists of two stages: Paraphrastic Probing, which identifies and extracts critical tokens, and Consistency Verification, which uses paraphrase forms to generate new reasoning trajectories with alternative tokens to reach the final answer. We evaluate our proposed framework with different LLMs and extensive evaluations across multiple benchmarks demonstrate the promising performance of our method.

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

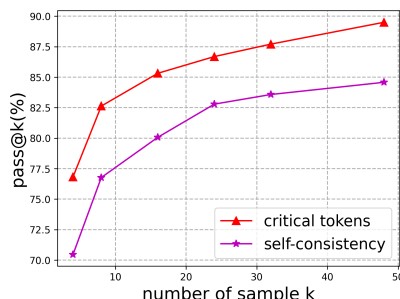

Figure 8: Comparison of the effects of critical tokens and Self-Consistency on the reasoning performance of LLMs, evaluated on samples from the GSM8K training data.

# A APPENDIX

## A.1 EMPIRICAL STUDY

We follow the heuristic criterion introduced in prior work (Lin et al., 2025) to evaluate the impact of critical tokens on the reasoning performance of LLMs and compare our approach with baselines such as Self-Consistency (Wang et al., 2023). In our empirical study, we sample multiple new rollouts at each token position along the reasoning path and evaluate their pass@k performance. A token is identified as critical if it satisfies two conditions: (1) the performance of its new rollouts is 0, and (2) the performance of subsequent tokens' rollouts falls below 5%. After identifying a critical token, we truncate the initial reasoning path at its position, replace it with alternative tokens, and then continue rolling out new reasoning paths. We compare the performance of these new rollouts against baselines such as Self-Consistency (Wang et al., 2023). The results on GSM8K (Cobbe et al., 2021), shown in Figure 8, highlight the pivotal role of critical tokens in shaping reasoning performance.

## A.2 AUTOMATIC PROMPT ENGINEERING FOR PARAPHRASING

The quality of paraphrased questions is crucial for our framework, as it directly affects both the identification of critical tokens and the stability of paraphrase-based consistency verification. To ensure high-quality paraphrasing, we adopt Automatic Prompt Engineering (APE) (Zhou et al., 2022) as an evaluation to optimize the paraphrasing instruction. This allows us to systematically control the quality of paraphrases rather than relying solely on manually written prompts. The APE procedure we employ consists of four steps:

- We create a small set of original questions paired with valid paraphrased variants. Using this exemplar set, the model generates multiple candidate instructions that could produce the paraphrased outputs from the original questions.
- Prompt each candidate instruction to the language model to generate paraphrases for the original problems and compare the mean solve rate change before and after paraphrasing.
- Choose the instruction that maximizes the mean solve rate change.
- Repeat the previous steps multiple times.

In our experiments, we initialize the paraphrasing instruction and iteratively refine it using APE with samples drawn from the GSM8K and MATH500 training sets. With this approach, the refined paraphrasing prompt helps us reliably produce high-quality paraphrased questions.

## A.3 MAIN RESULTS

To further assess the effectiveness of our method, we evaluate it on a larger reasoning model such as DeepSeek-R1-Distill-Llama-70B. We apply our full pipeline and compare against all baselines

Table 5: Comparison between our method and baselines with DeepSeek-R1-Distill-Llama-70B model on mathematical reasoning benchmarks.

| Method | AIME2024 | AIME2025 | BRUMO2025 | HMMT2025 |
|---|---|---|---|---|
| Chain-of-Thought | 56.67 | 38.00 | 43.33 | 30.00 |
| Predictive Decoding | 60.00 | 40.66 | 44.66 | 30.66 |
| Phi-Decoding | 64.00 | 46.67 | 48.00 | 31.33 |
| PPCV (Ours) | **70.00** | **56.66** | **56.66** | **33.33** |

Table 6: Comparison of our method and the baseline approach on Llama-3.1-8B-Instruct, evaluated using pass@k (k=4).

| Method | GSM8K | GSM-Hard | Math500 | SVAMP | ARC |
|---|---|---|---|---|---|
| Phi-Decoding | 92.15 | 53.57 | 52.60 | 91.19 | 90.73 |
| PPCV (Ours) | **93.83** | **61.41** | **59.39** | **94.48** | **94.24** |

Table 7: Comparison of our method and the baseline approach on Qwen3-32B, evaluated using pass@k (k=4).

| Method | AIME2024 | AIME2025 | BRUMO2025 | HMMT2025 |
|---|---|---|---|---|
| Phi-Decoding | 41.61 | 30.19 | 45.09 | 13.11 |
| PPCV (Ours) | **49.71** | **34.28** | **51.42** | **19.28** |

Table 8: The comparison of performance between Paraphrased Majority Voting (PMV) and our proposed PPCV.

| Method | GSM8K | GSM-Hard | Math500 | SVAMP | ARC |
|---|---|---|---|---|---|
| Chain-of-Thought | 77.40 | 28.00 | 31.00 | 83.00 | 58.91 |
| PMV | 78.55 | 30.16 | 32.60 | 84.10 | 60.63 |
| PPCV (Ours) | **88.24** | **49.73** | **50.00** | **89.60** | **88.31** |

across multiple mathematical reasoning benchmarks. The results, presented in Table 5, show that our method consistently outperforms the baselines. These improvements demonstrate that our approach remains effective for stronger reasoning models and generalizes well beyond the smaller models.

we also conduct additional experiments to report pass@k performance for both our method and the baselines. In these experiments, we use Phi-Decoding as the representative baseline and evaluate on two models: Llama-3.1-8B-Instruct and Qwen3-32B. We set k= 4 and generate 12 samples per question to obtain stable estimates of pass@4. The results across multiple benchmarks are presented in Table 6 and Table 7, respectively. Consistent with our main findings, our method achieves higher pass@k scores compared to the baseline methods, indicating that paraphrastic critical-token refinement continues to provide benefits in a multi-sample setting. These results further validate the robustness of our approach under stochastic sampling and demonstrate improved hit rates when multiple outputs are available.

## A.4 ABLATION STUDY

We also perform a comparison against a Paraphrased Majority Voting (PMV) strategy is essential to demonstrate that the performance gains of our method are indeed derived from the critical token correction, rather than simply the benefit of having multiple paraphrased samples. Thus, we have performed the suggested additional experiment. We adopt Paraphrased Majority Voting (PMV) as the baseline. This approach uses the same total number of trajectories generated from the paraphrased questions as PPCV, and reaches the final answer using standard majority voting over the rollouts from the paraphrased questions. We compare the PMV baseline against our full PPCV method and ensure the same number of paraphrased questions as in our method. The results shown in Table 8 demonstrate that the performance yielded by PMV is significantly lower than that of our proposed PPCV method. This result confirms that simply rolling out trajectories from paraphrased questions and aggregating the answers is insufficient. The substantial performance gap between PMV and PPCV decisively highlights the necessity and effectiveness of our extracted critical tokens and the subsequent corrective reasoning step.

A.5    PROMPTS

---

**Prompts for reasoning** (for math datasets)

System: You are a specialized AI assistant designed to solve mathematical problems.

User: Please solve the following problem step by step, and put your final answer within boxed{}.

Each pack of dvds costs 76 dollars. If there is a discount of 25 dollars on each pack How much do you have to pay to buy each pack?

Assistant: To find the price after the discount, we need to subtract the discount amount from the original price.
Step 1: Identify the original price and the discount amount.Original price = $76. Discount amount = $25
Step 2: Subtract the discount amount from the original price. Price after discount = Original price - Discount amount. Price after discount = $76 - $25. Price after discount = $51
So, you have to pay $51 to buy each pack.
boxed{$51}

---

**Prompts for reasoning** (for ARC dataset)

System: You are a specialized AI assistant designed to solve complex reasoning problems.

User: Your task is to carefully read the following question, apply logical reasoning, scientific knowledge, and commonsense inference, and arrive at the correct answer. Always explain your reasoning step by step in clear language. At the end of your response, output the final choice in the following format: The answer is: (option), where option is the correct option letter (A, B, C, D, etc.), enclosed in parentheses.

A group of engineers wanted to know how different building designs would respond during an earthquake. They made several models of buildings and tested each for its ability to withstand earthquake conditions. Which will most likely result from testing different building designs?
(A) buildings will be built faster
(B) buildings will be made safer
(C) building designs will look nicer
(D) building materials will be cheaper?

Assistant: We are given a scenario in which engineers are testing different building designs to see how well they withstand earthquake conditions. The goal is to determine which of the listed outcomes is most likely to result from such testing.
Let 2019s evaluate each option in turn:
(A) Buildings will be built faster. 2013 This is not directly related to how well a building withstands earthquakes. Earthquake resistance is about structural integrity, not construction speed. While a good design might simplify construction, the primary purpose of the test is not speed. So this is not the most likely result.
(B) Buildings will be made safer 2013 This is directly aligned with the purpose of the experiment. Testing building designs under earthquake conditions helps identify which designs can better absorb or resist seismic forces. The goal is to improve safety during earthquakes. Therefore, this outcome is a direct and logical result of such testing.
(C) Building designs will look nicer 2013 Aesthetics are not the focus of earthquake testing. While a design might be both safe and attractive, the test itself does not evaluate visual appeal. So this is not a likely result of the testing.
(D) Building materials will be cheaper 2013 Cost of materials is not necessarily a result of testing for earthquake resistance. ...
The answer is: (B)

---

**Prompts to paraphrase the question** (for math datasets)

System: You are a paraphrasing assistant specialized in math problems. Your task is to paraphrase a given math word problem. The core objective is to create a new, diverse expression of the problem while strictly preserving all numerical values, mathematical relationships, and the core logic required to solve it.

User: Please create 5 paraphrased versions of the question below. Do NOT change any numbers or quantities. Maintain the exact mathematical operations and relationships. And vary the context and language. Please output paraphrased questions starting with a bracketed serial number like [1] [2] and ending with a new line.

A robe takes 2 bolts of blue fiber and half that much white fiber. How many bolts in total does it take?

Assistant:
[1] Two bolts of blue fabric and one and a half times that amount of white fabric are required to make a robe. How many bolts in all are needed?

[2] A robe is made from 2 bolts of blue yarn and an additional amount equal to half of that for white yarn. In total, how many bolts does it call for?

[3] To create a robe, you'll need 2 bolts of blue thread and another half-quantity of white thread. How many bolts of thread in total should you purchase?

[4] For a single robe, the production process demands 2 bolts of blue fabric and 1.5 times as much white fabric. What is the total number of bolts required?

[5] The manufacture of a robe necessitates the usage of 2 bolts of blue material and slightly less than double that quantity for white material. How many total bolts does this equate to?

---

**Prompts to paraphrase the question** (for ARC dataset)

System: You are given a multiple-choice question from a reasoning benchmark. Your task is to generate multiple different paraphrased versions of the question. Each version must preserve the original meaning, answer choices, and scientific/commonsense accuracy.
Guidelines: - Keep the question logically equivalent to the original. - Do not change the correct answer. - Do not add or remove details that would alter the difficulty. - Use different phrasing, sentence structures, or synonyms for each version. - Keep the multiple-choice options (A, B, C, D, etc.) unchanged.
Example Input: A large gene has dozens of exons, of which the central ones code for folded triple helical repeats that connect the cytoskeleton with sarcolemma and extracellular space. Each exon usually codes for one folded triple alpha helix. The most common mutations of the gene are central exon deletions that create out-of-frame peptides and progressive degenerative organ waste. A solution is to deliver a Morpholino that recognizes the 5' end of the out-of-frame exon in pre-mRNA. The molecule prevents binding of the spliceosome and creates exon skipping and in-frame joining. Several missing exons are well tolerated by an organism. Which structure below is not involved in the proposed therapy?

(A) antisense (B) lariat (C) R-loops (D) polyA tail

User: Please create 3 paraphrased versions of the question below. Only paraphrases the question and leaves the multiple-choice options untouched. Please output paraphrased questions starting with a bracketed serial number like [1] [2] and ending with a new line,

An astronomer observes that a planet rotates faster after a meteorite impact. Which is the most likely effect of this increase in rotation?
(A) Planetary density will decrease.
(B) Planetary years will become longer.
(C) Planetary days will become shorter.
(D) Planetary gravity will become stronger.

Assistant:
[1] An astronomer notices that a planet's rotational speed increases following a meteorite collision. What is the most probable consequence of this accelerated rotation? (A) Planetary density will decrease.
(B) Planetary years will become longer.
(C) Planetary days will become shorter.
(D) Planetary gravity will become stronger.

[2] After a meteorite strikes a planet, the astronomer observes an increase in its rotational velocity. What is the most likely outcome of this faster rotation? (A) Planetary density will decrease.
(B) Planetary years will become longer.
(C) Planetary days will become shorter.
(D) Planetary gravity will become stronger.

[3] A planet begins to rotate more quickly after experiencing a meteorite impact. Which of the following is most likely to result from this increased spin? (A) Planetary density will decrease.
(B) Planetary years will become longer.
(C) Planetary days will become shorter.
(D) Planetary gravity will become stronger.

---

## A.6 USE OF LARGE LANGUAGE MODELS

In our work, we focus on improving large Malaguena models, and LLMs are used in our experimental evaluation. In addition, we employ LLMs to check grammatical errors and refine sentence structures.

