# OpenReview forum: "Finding the Cracks: Improving LLMs Reasoning with Paraphrastic Probing and Consistency Verification"
_ICLR.cc/2026/Conference — Submitted to ICLR 2026_

### Official Review · Reviewer_s9UP · 2025-10-23

**Soundness:** 3
**Presentation:** 3
**Contribution:** 3
**Rating:** 6
**Confidence:** 4

**Summary:**

The paper proposes a new inference time optimization method: paraphrastic probing and consistency verification. The main idea is (1) identify the critical tokens (tokens that may trigger very different reasoning paths) by running inference on paraphrased inputs + original response; (2) generate new rollouts by replacing the critical tokens in the original response and finally do a majority vote on all responses.
The empirical results show that the proposed method outperforms previous inference-time scaling methods such as self-consistency, across reasoning-intensive tasks such as Math and ARC-challenge.

**Strengths:**

- The paper is well-written. The method is clearly explained and easy to follow.
- The experimental results and ablation supports most of the paper claims, and shows the importance of both the critical tokens and the paraphrased consistency.

**Weaknesses:**

- One important baseline is still missing: simply perform majority voting but on the paraphrased questions (be sure to keep the total number instances the same as the PPCV method) => this will show that whether the rollout based on critical tokens really helps. Although the authors provide the comparison with random tokens, it does not make sense, since replacing random tokens may artificially shift the model's distribution too much and hurt it's original performance.
- The computational analysis is too brief and lacks important details: I am suspicious about the correctness of the inference time figure: in most datasets, PPCV uses only about 2x the inference time than simple CoT, this does not seem to be possible. Becuase the computation overhead of PPCV comes from (1) N forward pass on q1...qN with the orignal question; (2) K * N generation sequences for each critical token and each question variant. Particularly, suppose the critical token is on average in the middle of the response, the stage (2) should add approximately 0.5 * K * N times of the original CoT generation time (regardless of the prefilling time).

**Questions:**

- The fonts in most figures are too small, e.g., Figure 3,4,5; would be good to enlarge

---

> ### Author Response · Authors · 2025-11-24
> **Response 1**
>
> **W1:** One important baseline is still missing: simply perform majority voting but on the paraphrased questions (be sure to keep the total number instances the same as the PPCV method) => this will show that whether the rollout based on critical tokens really helps. Although the authors provide the comparison with random tokens, it does not make sense, since replacing random tokens may artificially shift the model's distribution too much and hurt it's original performance.
>
> **R1:** We thank the reviewer for proposing this important baseline. We agree that a comparison against a Paraphrased Majority Voting (PMV) strategy is essential to demonstrate that the performance gains of our method are indeed derived from the critical token correction, rather than simply the benefit of having multiple paraphrased samples. Thus, we have performed the suggested additional experiment. We adopt Paraphrased Majority Voting (PMV) as the baseline. This approach uses the same total number of trajectories generated from the paraphrased questions as PPCV, and reaches the final answer using standard majority voting over the rollouts from the paraphrased questions. We compare the PMV baseline against our full PPCV method and ensure the same number of paraphrased questions as in our method. The results shown in Table 1 demonstrate that the performance yielded by PMV is significantly lower than that of our proposed PPCV method. This result confirms that simply rolling out trajectories from paraphrased questions and aggregating the answers is insufficient. The substantial performance gap between PMV and PPCV decisively highlights the necessity and effectiveness of our extracted critical tokens and the subsequent corrective reasoning step. The results and relevant discussion have been added to our revised paper.
>
>  **Table 1. The comparison of performance between Paraphrased Majority Voting (PMV) and our proposed PPCV.**
>
> | Method        | GSM8K | GSM-Hard | Math500 | SVAMP | ARC   |
> |---------------|-------|----------|---------|-------|-------|
> | PMV           | 78.55 | 30.16    | 32.60   | 84.10 | 60.63 |
> | PPCV (Ours)   | 88.24 | 49.73    | 50.00   | 89.60 | 88.31 |

---

> ### Author Response · Authors · 2025-11-24
> **Response 2**
>
> **W2.** The computational analysis is too brief and lacks important details.
>
> **R2:** Thank you for pointing out this issue. Upon review, we found that our previous computational cost analysis was flawed and lacked sufficient details. As a result, the reported results did not accurately reflect the true computational overhead of our method and may have caused confusion. We have therefore carefully reexamined the computational cost of our approach.
>
> We first examine the composition of the latency in our method. The overall latency arises from four components:
> 1. Paraphrased question generation (T1).
> 2. Initial answer generation (T2​), equivalent to vanilla CoT.
> 3. Forward pass for identifying critical tokens (T3​), which does not generate new tokens and is computationally lightweight.
> 4. Rollouts of truncated trajectories using alternative tokens under both the original and paraphrased questions (T4​), which constitutes the main source of overhead.
>
> We have evaluated all components on Llama-3.1-8B-Instruct using vLLM on NVIDIA A100 GPUs, with a maximum output length of 4096 tokens for each question. For our method, we use 5 paraphrased questions on math datasets and 3 on ARC, and select the top-10 candidate tokens as alternatives. The updated average latency results are reported in Table 2. As expected, T1​ scales with the number of paraphrases, T3​ remains minimal, and T4 dominates the total cost. Specifically, T4 depends on the number of top-k alternative tokens, the number of paraphrased questions and the position of the critical token in the trajectory. Since the new rollouts from truncated trajectories for different alternative tokens and paraphrased questions are independent, T4 ​can take advantage of vLLM’s parallelism. These rollouts can therefore be processed concurrently, improving overall efficiency. This is reflected in the higher throughput (tokens/sec) shown in Table 3. And results of our method in latency comparable to baseline methods, even on challenging benchmarks such as MATH500 and GSM-Hard where the critical token tends to occur in later reasoning steps. On benchmarks such as GSM8K and SVAMP, both our method and baselines like Predictive Decoding exhibit higher latency compared to vanilla Chain-of-Thought.
>
> We have also conducted a trade-off analysis between performance and efficiency by varying the number of alternative tokens considered at the critical position. Specifically, we test top-3, top-5, top-7, and top-10 alternatives using GSM8K, sampling 200 random questions and repeating the experiment five times to compute confidence intervals. The results are presented in Table 4. We observe a clear trade-off. Reducing the number of alternative tokens lowers both latency and throughput, while causing a slight reduction in accuracy. This provides a practical mechanism for adjusting performance-efficiency trade-offs in real deployments. Depending on resource availability and target accuracy, practitioners can choose the appropriate number of alternative tokens. The new results and relevant discussion have been incorporated in our revised paper.
>
>
>  **Table 2. Latency comparison between the baselines and our method, measured as the average inference time per question (in seconds).**
>
> | Dataset   | CoT  | Predive Dec | Phi-Dec |  PPCV(T1)   | PPCV(T2)   | PPCV(T3)   | PPCV(T4)    | PPCV(Total) |
> |-----------|------|----------|---------|------|------|------|-------|-------|
> | GSM8K     | 2.11 | 15.43    | 12.99   | 2.40 | 2.11 | 0.30 | 12.94 | 17.75 |
> | GSM-Hard  | 3.01 | 26.20    | 23.25   | 2.91 | 3.01 | 0.40 | 16.59 | 22.91 |
> | Math500   | 7.29 | 42.37    | 37.90   | 2.50 | 7.29 | 0.56 | 25.03 | 35.35 |
> | SVAMP     | 1.79 | 14.18    | 11.17   | 1.91 | 1.79 | 0.31 | 12.67 | 16.68 |
> | ARC       | 2.14 | 15.60    | 15.40   | 1.33 | 2.14 | 0.21 | 4.72  | 8.40  |
>
>  **Table 3. Throughput comparison between the baselines and our method, measured in output tokens per second.**
>
> | Method              | GSM8K    | GSM-Hard | Math500  | SVAMP    | ARC      |
> |---------------------|----------|----------|----------|----------|----------|
> | Chain-of-Thought    | 121.19   | 128.95   | 133.85   | 112.94   | 128.71   |
> | Predictive Decoding | 691.52   | 600.23   | 791.02   | 539.18   | 758.62   |
> | Phi-Decoding        | 507.94   | 436.49   | 567.68   | 397.17   | 584.80   |
> | PPCV (ours)         | 1308.66  | 1707.86  | 2028.82  | 1526.61  | 1503.15  |
>
>  **Table 4. Comparison of our method under different numbers of top-k tokens (k = 3, 5, 7, 10). Reported metrics include latency (s), throughput (tokens/s), and accuracy (%).**
>
> | Top-k | Latency      | Throughput     | Accuracy     |
> |-------|--------------|----------------|--------------|
> | Top-3 | 14.08±0.56   | 547.67±13.10   | 86.43±1.21   |
> | Top-5 | 15.86±0.48   | 796.05±16.98   | 87.88±1.23   |
> | Top-7 | 17.01±0.64   | 1011.89±39.92  | 88.43±1.54   |
> | Top-10| 17.94±0.30   | 1332.07±60.05  | 88.68±1.60   |

---

> ### Author Response · Authors · 2025-11-24
> **Resposne 3**
>
> **Q1:The fonts in most figures are too small, e.g., Figure 3,4,5; would be good to enlarge.**
>
> **R3:** Thank you for your suggestions. We have increased the font sizes in the corresponding figures in the revised version of the paper.

---

### Official Review · Reviewer_9jC6 · 2025-10-26

**Soundness:** 3
**Presentation:** 2
**Contribution:** 2
**Rating:** 4
**Confidence:** 4

**Summary:**

This paper aims to improve the reasoning performance of LLMs by proposing a Paraphrastic Probing and Consistency Verification (PPCV) framework. Specifically, PPCV consists of two stages: 1) roll out the initial reasoning path from the original question, then concatenate paraphrased questions with the reasoning path, which is fed into the LLM to identify the critical tokens; 2) leverage the extracted critical tokens to refine the initial reasoning path via a self-consistency mechanism. Experiments on multiple mathematics and logical reasoning benchmarks show that PPCV outperforms the vanilla CoT and some baseline decoding methods.

**Strengths:**

-	The paper is overall well-written and easy to follow.
-	The idea of locating the critical tokens makes sense, and does not rely on external models.
-	Experimental results show that PPCV outperforms the other counterparts by a clear margin.

**Weaknesses:**

-	My main concern is the efficiency of the proposed framework. If I understand correctly, PPCV requires multiple passes of LLMs, e.g., paraphrasing the question, obtaining the initial reasoning paths, obtaining the critical tokens by feeding into multiple paraphrased questions, and generating a group of new trajectories. The total framework is complex and would lead to much inference latency. The comparison of the inference time in Figure 7 is also confusing. For example, in the case of the SVAMP test set, PPCV only brings a slight inference latency against the vanilla CoT method, which is obviously counterintuitive. The authors should provide more explanation.
-	The technical contribution is limited. In my opinion, PPCV does not propose new technologies, but uses existing methods and ideas to solve a relatively new problem.
-	In the ablation study, the authors do not analyze the influence of paraphrased questions. For example, remove the paraphrase processes in PPCV and explore the performance changes.
-	Algorithm 1 is a little hard to paraphrase. It will be better to improve the presentation carefully.

**Questions:**

See the Weaknesses.

---

> ### Author Response · Authors · 2025-11-24
> **Response 1**
>
> **W1: My main concern is the efficiency of the proposed framework.**
>
> **R1:** Thank you for pointing out this issue. Upon review, we found that our previous computational cost analysis was flawed and lacked sufficient details. As a result, the reported results did not accurately reflect the true computational overhead of our method and may have caused confusion. We have therefore carefully reexamined the computational cost of our approach.
>
> We first examine the composition of the latency in our method. The overall latency arises from four components:
> 1. Paraphrased question generation (T1).
> 2. Initial answer generation (T2​), equivalent to vanilla CoT.
> 3. Forward pass for identifying critical tokens (T3​), which does not generate new tokens and is computationally lightweight.
> 4. Rollouts of truncated trajectories using alternative tokens under both the original and paraphrased questions (T4​), which constitutes the main source of overhead.
>
> We have evaluated all components on Llama-3.1-8B-Instruct using vLLM on NVIDIA A100 GPUs, with a maximum output length of 4096 tokens for each question. For our method, we use 5 paraphrased questions on math datasets and 3 on ARC, and select the top-10 candidate tokens as alternatives. The updated average latency results are reported in Table 1. As expected, T1​ scales with the number of paraphrases, T3​ remains minimal, and T4 dominates the total cost. Specifically, T4 depends on the number of top-k alternative tokens, the number of paraphrased questions and the position of the critical token in the trajectory. Since the new rollouts from truncated trajectories for different alternative tokens and paraphrased questions are independent, T4 ​can take advantage of vLLM’s parallelism. These rollouts can therefore be processed concurrently, improving overall efficiency. This is reflected in the higher throughput (tokens/sec) shown in Table 2. And results of our method in latency comparable to baseline methods, even on challenging benchmarks such as MATH500 and GSM-Hard where the critical token tends to occur in later reasoning steps. On benchmarks such as GSM8K and SVAMP, both our method and baselines like Predictive Decoding exhibit higher latency compared to vanilla Chain-of-Thought.
>
> We have also conducted a trade-off analysis between performance and efficiency by varying the number of alternative tokens considered at the critical position. Specifically, we test top-3, top-5, top-7, and top-10 alternatives using GSM8K, sampling 200 random questions and repeating the experiment five times to compute confidence intervals. The results are presented in Table 3. We observe a clear trade-off. Reducing the number of alternative tokens lowers both latency and throughput, while causing a slight reduction in accuracy. This provides a practical mechanism for adjusting performance-efficiency trade-offs in real deployments. Depending on resource availability and target accuracy, practitioners can choose the appropriate number of alternative tokens. The new results and relevant discussion have been incorporated in our revised paper.
>
>
>  **Table 1. Latency comparison between the baselines and our method, measured as the average inference time per question (in seconds).**
>
> | Dataset   | CoT  | Predive Dec | Phi-Dec |  PPCV(T1)   | PPCV(T2)   | PPCV(T3)   | PPCV(T4)    | PPCV(Total) |
> |-----------|------|----------|---------|------|------|------|-------|-------|
> | GSM8K     | 2.11 | 15.43    | 12.99   | 2.40 | 2.11 | 0.30 | 12.94 | 17.75 |
> | GSM-Hard  | 3.01 | 26.20    | 23.25   | 2.91 | 3.01 | 0.40 | 16.59 | 22.91 |
> | Math500   | 7.29 | 42.37    | 37.90   | 2.50 | 7.29 | 0.56 | 25.03 | 35.35 |
> | SVAMP     | 1.79 | 14.18    | 11.17   | 1.91 | 1.79 | 0.31 | 12.67 | 16.68 |
> | ARC       | 2.14 | 15.60    | 15.40   | 1.33 | 2.14 | 0.21 | 4.72  | 8.40  |
>
>  **Table 2. Throughput comparison between the baselines and our method, measured in output tokens per second.**
>
> | Method              | GSM8K    | GSM-Hard | Math500  | SVAMP    | ARC      |
> |---------------------|----------|----------|----------|----------|----------|
> | Chain-of-Thought    | 121.19   | 128.95   | 133.85   | 112.94   | 128.71   |
> | Predictive Decoding | 691.52   | 600.23   | 791.02   | 539.18   | 758.62   |
> | Phi-Decoding        | 507.94   | 436.49   | 567.68   | 397.17   | 584.80   |
> | PPCV (ours)         | 1308.66  | 1707.86  | 2028.82  | 1526.61  | 1503.15  |
>
>  **Table 3. Comparison of our method under different numbers of top-k tokens (k = 3, 5, 7, 10). Reported metrics include latency (s), throughput (tokens/s), and accuracy (%).**
>
> | Top-k | Latency      | Throughput     | Accuracy     |
> |-------|--------------|----------------|--------------|
> | Top-3 | 14.08±0.56   | 547.67±13.10   | 86.43±1.21   |
> | Top-5 | 15.86±0.48   | 796.05±16.98   | 87.88±1.23   |
> | Top-7 | 17.01±0.64   | 1011.89±39.92  | 88.43±1.54   |
> | Top-10| 17.94±0.30   | 1332.07±60.05  | 88.68±1.60   |

---

> ### Author Response · Authors · 2025-11-24
> **Response 2**
>
> **W2: The technical contribution is limited. In my opinion, PPCV does not propose new technologies, but uses existing methods and ideas to solve a relatively new problem.**
>
> **R2:** We thank the reviewer for this assessment and the opportunity to clarify our technical contributions. While we agree that paraphrasing and sampling are existing concepts, our work proposes a novel framework that repurposes these concepts in a fundamentally new way to address the unreliable reasoning of LLMs.
>
> Our technical contributions differ from prior works in three distinct ways. First, prior works [1-3] typically use paraphrasing merely to expand the solution space. In contrast, we introduce Paraphrastic Probing, a mechanism that uses paraphrasing to test the model's internal confidence. By analyzing the discrepancy in token-level logits of the initial trajectory between the original and paraphrased questions, we can rigorously pinpoint the critical tokens that may lead to errors in the following steps.This transforms paraphrasing from a generation tool into a precise, token-level diagnostic tool.
>
> Second, prior works [1-3] typically rely on simple majority voting across multiple solutions. Our Paraphrase Consistency mechanism is technically distinct. It validates answers based on their robustness across semantic variations of the problem constraint. We further introduce a similarity-weighted consistency metric that weighs answers based on the linguistic diversity of the paraphrase, offering a more nuanced selection criterion than simple frequency counts.
>
> At last, a major technical limitation in current reasoning research is the reliance on external Process Reward Models (PRMs) or human-annotated error steps. PPCV contributes a fully self-contained pipeline that identifies and corrects errors using the model's own sensitivity to surface-form perturbations. We believe our proposed framework represents a significant methodological advancement, offering a way to "find the cracks" in reasoning without the heavy computational or data overhead of training separate verifiers. The relevant discussion has been added to our revised paper.
>
> [1] Zhou, Yue, et al. "Paraphrase and Solve: Exploring and Exploiting the Impact of Surface Form on Mathematical Reasoning in Large Language Models." NAACL (Volume 1: Long Papers). 2024.
>
> [2] Chen, Wenqing, et al. "Self-Para-Consistency: Improving Reasoning Tasks at Low Cost for Large Language Models." Findings of ACL 2024.
>
> [3] Yadav, Vikas, Zheng Tang, and Vijay Srinivasan. "Pag-llm: Paraphrase and aggregate with large language models for minimizing intent classification errors." ACM SIGIR. 2024.

---

> ### Author Response · Authors · 2025-11-24
> **Response 3**
>
> **W3:** In the ablation study, the authors do not analyze the influence of paraphrased questions. For example, remove the paraphrase processes in PPCV and explore the performance changes.
>
> **R3:** We thank the reviewer for the suggestion. We agree that isolating the impact of paraphrasing is crucial. In our manuscript, we performed this exact analysis by decoupling the two stages of our framework: Paraphrastic Probing (Phase 1) and Paraphrase Consistency (Phase 2). The influence of paraphrasing in each phase has already been analyzed separately in Section 4.3 ("Ablation Study").
>
> First, we investigate the influence of paraphrasing in Phase 1 for the probing purpose.  If we remove the paraphrase process from Phase 1, the model loses the ability to identify critical tokens based on logit discrepancies between original and paraphrased contexts. Consequently, tokens would be selected without the guidance. We simulated this "no-paraphrase" condition by selecting random tokens instead of critical tokens. The results shown in Table 4 suggest removing the paraphrastic probing mechanism leads to a substantial decline in performance. This confirms that the paraphrasing process in Phase 1 is essential for locating error-prone steps.
>
> Next, we investigate the influence of paraphrasing in Phase 2 for the verification purpose: If we remove the paraphrase process from Phase 2, we can no longer verify consistency across different problem formulations. Instead, the model must rely on standard consistency checks on the original question alone. This would revert the method to traditional majority voting. The results shown in Table 5 indicate that removing the paraphrase consistency from the verification stage results in lower accuracy across all benchmarks. This demonstrates that paraphrasing in Phase 2 provides a more robust verification signal than simple majority voting.
>
>  **Table 4. Comparison of our method in Phase 1 (paraphrastic probing) with and without the paraphrasing process.**
>
> | Method                  | GSM8K | GSM-Hard | Math500 | SVAMP | ARC   |
> |-------------------------|-------|----------|---------|-------|-------|
> | Phase 1 (without Paraphrase) | 82.08 | 40.29    | 42.12   | 84.77 | 75.68 |
> | Phase 1 (with Paraphrase)    | 88.24 | 49.73    | 50.00   | 89.60 | 88.31 |
>
>  **Table 5. Comparison of our method in Phase 2 (paraphrase consistency) with and without the paraphrasing process.**
>
> | Method                  | GSM8K | GSM-Hard | Math500 | SVAMP | ARC   |
> |-------------------------|-------|----------|---------|-------|-------|
> | Phase 2 (without Paraphrase) | 87.20 | 47.36    | 48.19   | 88.80 | 86.16 |
> | Phase 2 (with Paraphrase)    | 88.24 | 49.73    | 50.00   | 89.60 | 88.31 |
>
> **W4:** Algorithm 1 is a little hard to paraphrase. It will be better to improve the presentation carefully.
>
> **R4:**  We appreciate this valuable feedback. We have carefully revised Algorithm 1 to enhance its clarity and readability.The updated algorithm has been incorporated into the revised paper.

---

### Official Review · Reviewer_MkiR · 2025-11-01

**Soundness:** 3
**Presentation:** 3
**Contribution:** 3
**Rating:** 4
**Confidence:** 5

**Summary:**

This paper introduced PPCV, a framework that leverages critical tokens to improve reasoning performance. First, the framework roll out an initial reasoning path from the original question and then concatenate paraphrased versions of the question with this reasoning path. Everything is passed to the LLM to get token-level logits to identify critical tokens based on mismatches between the predicted top-1 token and the expected token in the reasoning path. In the second stage, critical tokens are substituted with candidate alternatives and the framework rolls out new reasoning paths for both the original and paraphrased questions. The final answer is determined by checking the consistency of outputs across these parallel reasoning processes.

First, the authors conduct an empirical analysis of the effectiveness of critical tokens: identify the one critical token, replace it. Compare performance with self-consistency. Indeed, critical token replacements is more beneficial.

In terms of methodology, an LLM is first prompted to paraphrase a problem multiple times while keeping all numerical values etc the same. Then a reasoning path is computed with the original question. The reasoning path is concatenated to each paraphrased version. Then the top-1 log-probs are computed for each token. The first one that does not match the the top-1 on the original problem is considered critical.
To refine the original reasoning path with alternative tokens, an LLM is prompted from the beginning until the critical token. A consistency score is computed on all alternatives.

Regarding the experiments, Llama 8B, Mistral 7B, and Qwen 3 32b are used on multiple reasoning datasets. Baseline consists of CoT, ToT, guided decoding, predictive decoding and phi-decoding. Each dataset is paraphrased 3-5 times. PPVC seems to lead to the highest performance on 7B thinking models and Qwen3 32B. I would encourage the authors to compare higher reasoning models and use pass@k. The ablations are well done. Finally, I'm a bit disappointed by the computational cost analysis with Figure 7. There is no confidence interval, variation, and it is hard to weight the trade-off between performance and efficiency. Could you provide a Pareto plot with performance vs throughput and include confidence interval?

**Strengths:**

- Interesting idea.
- Good ablation analysis.

**Weaknesses:**

- No analysis with larger reasoning models.
- No use of pass@k.
- Computational cost analysis is insufficient.

**Questions:**

- Could you compare performance on larger reasoning models?
- Could you report pass@k for Table1 and Table2 (k>= 4)?
- In Table 3&ç, please add the original performance.

---

> ### Author Response · Authors · 2025-11-24
> **Response 1**
>
> **W1, Q1: No analysis with larger reasoning models.**
>
> **R1:** Thank you for this valuable suggestion. To further assess the effectiveness of our method, we evaluate it on a larger reasoning model such as DeepSeek-R1-Distill-Llama-70B. We apply our full pipeline and compare against all baselines across multiple mathematical reasoning benchmarks. The results, presented in Table 1 below, show that our method consistently outperforms the baselines. These improvements demonstrate that our approach remains effective for stronger reasoning models and generalizes well beyond the smaller-scale models reported in the original manuscript. We have added the new results and discussions in our revised paper.
>
> **Table 1. Comparison between our method and baselines with DeepSeek-R1-Distill-Llama-70B on mathematical reasoning benchmarks.**
>
> | Method             | AIME2024 | AIME2025 | BRUMO2025 | HMMT2025 |
> |--------------------|----------|----------|-----------|----------|
> | Chain-of-Thought   | 56.67    | 38.00    | 43.33     | 30.00    |
> | Predictive Decoding| 60.00    | 40.66    | 44.66     | 30.66    |
> | Phi-Decoding       | 64.00    | 46.67    | 48.00     | 31.33    |
> | PPCV (Ours)        | **70.00**    | **56.66**    | **56.66**     | **33.33**    |
>
> **W2, Q2: No use of pass@k.**
>
> **R2:** Thank you for the helpful suggestion. Following your advice, we have conducted additional experiments to report pass@k performance for both our method and the baselines. In these experiments, we use Phi-Decoding as the representative baseline and evaluate on two models: Llama-3.1-8B-Instruct and Qwen3-32B. We set k= 4 and generate 12 samples per question to obtain stable estimates of pass@4.
> The results across multiple benchmarks are presented in Table 2 and Table 3 below. Consistent with our main findings, our method achieves higher pass@k scores compared to the baseline methods, indicating that paraphrastic critical-token refinement continues to provide benefits in a multi-sample setting. These results further validate the robustness of our approach under stochastic sampling and demonstrate improved hit rates when multiple outputs are available. The new results and relevant discussion have been added to the revised paper.
>
> **Table 2. Comparison of our method and the baseline approach on Llama-3.1-8B-Instruct, evaluated using pass@k (k=4).**
>
> | Method        | GSM8K | GSM-Hard | Math500 | SVAMP | ARC   |
> |---------------|-------|----------|---------|-------|-------|
> | Phi-Decoding  | 92.15 | 53.57    | 52.60   | 91.19 | 90.73 |
> | PPCV (Ours)   | **93.83** | **61.41** | **59.39** | **94.48** | **94.24** |
>
> **Table 3. Comparison of our method and the baseline approach on Qwen3-32B, evaluated using pass@k (k=4).**
>
> | Method        | AIME2024 | AIME2025 | BRUMO2025 | HMMT2025 |
> |---------------|----------|----------|-----------|----------|
> | Phi-Decoding  | 41.61    | 30.19    | 45.09     | 13.11    |
> | PPCV (Ours)   | **49.71** | **34.28** | **51.42** | **19.28** |

---

> ### Author Response · Authors · 2025-11-24
> **Response 2**
>
> **W3: Computational cost analysis is insufficient.**
>
> **R3:** Thank you for pointing out the limitations in our previous computational cost analysis. We agree that the earlier version lacked necessary detail and did not sufficiently highlight the trade-off between performance and efficiency. We have therefore revisited and substantially expanded the analysis to provide a clearer and more rigorous evaluation.
> We first examine the composition of the latency in our method. The overall latency arises from four components:
> 1. Paraphrased question generation (T1).
> 2. Initial answer generation (T2​), equivalent to vanilla CoT.
> 3. Forward pass for identifying critical tokens (T3​), which does not generate new tokens and is computationally lightweight.
> 4. Rollouts of truncated trajectories using alternative tokens under both the original and paraphrased questions (T4​), which constitutes the main source of overhead.
>
> We have evaluated all components on Llama-3.1-8B-Instruct using vLLM on NVIDIA A100 GPUs, with a maximum output length of 4,096 tokens for each question. For our method, we use 5 paraphrased questions on math datasets and 3 on ARC, and select the top-10 candidate tokens as alternatives. The updated average latency results are reported in Table 4. As expected, T1​ scales with the number of paraphrases, T3​ remains minimal, and T4 dominates the total cost. Specifically, T4 depends on the number of top-k alternative tokens, the number of paraphrased questions and the position of the critical token in the trajectory. Since the new rollouts from truncated trajectories for different alternative tokens and paraphrased questions are independent, T4 ​can take advantage of vLLM’s parallelism. These rollouts can therefore be processed concurrently, improving overall efficiency. This is reflected in the higher throughput (tokens/sec) shown in Table 5. And results of our method in latency comparable to baseline methods, even on challenging benchmarks such as MATH500 and GSM-Hard where the critical token tends to occur in later reasoning steps.
>
> We have also conducted a trade-off analysis between performance and efficiency by varying the number of alternative tokens considered at the critical position. Specifically, we test top-3, top-5, top-7, and top-10 alternatives using GSM8K, sampling 200 random questions and repeating the experiment five times to compute confidence intervals. The results are presented in Table 6. We observe a clear trade-off. Reducing the number of alternative tokens lowers both latency and throughput, while causing a slight reduction in accuracy. This provides a practical mechanism for adjusting performance-efficiency trade-offs in real deployments. Depending on resource availability and target accuracy, practitioners can choose the appropriate number of alternative tokens. The new results and relevant discussion have been incorporated in our revised paper.
>
>  **Table 4. Latency comparison between the baselines and our method, measured as the average inference time per question (in seconds).**
>
> | Dataset   | CoT  | Predive Dec | Phi-Dec |  PPCV(T1)   | PPCV(T2)   | PPCV(T3)   | PPCV(T4)    | PPCV(Total) |
> |-----------|------|----------|---------|------|------|------|-------|-------|
> | GSM8K     | 2.11 | 15.43    | 12.99   | 2.40 | 2.11 | 0.30 | 12.94 | 17.75 |
> | GSM-Hard  | 3.01 | 26.20    | 23.25   | 2.91 | 3.01 | 0.40 | 16.59 | 22.91 |
> | Math500   | 7.29 | 42.37    | 37.90   | 2.50 | 7.29 | 0.56 | 25.03 | 35.35 |
> | SVAMP     | 1.79 | 14.18    | 11.17   | 1.91 | 1.79 | 0.31 | 12.67 | 16.68 |
> | ARC       | 2.14 | 15.60    | 15.40   | 1.33 | 2.14 | 0.21 | 4.72  | 8.40  |
>
>  **Table 5. Throughput comparison between the baselines and our method, measured in output tokens per second.**
>
> | Method              | GSM8K    | GSM-Hard | Math500  | SVAMP    | ARC      |
> |---------------------|----------|----------|----------|----------|----------|
> | Chain-of-Thought    | 121.19   | 128.95   | 133.85   | 112.94   | 128.71   |
> | Predictive Decoding | 691.52   | 600.23   | 791.02   | 539.18   | 758.62   |
> | Phi-Decoding        | 507.94   | 436.49   | 567.68   | 397.17   | 584.80   |
> | PPCV (ours)         | 1308.66  | 1707.86  | 2028.82  | 1526.61  | 1503.15  |
>
>  **Table 6. Comparison of our method under different numbers of top-k tokens (k = 3, 5, 7, 10). Reported metrics include latency (s), throughput (tokens/s), and accuracy (%).**
>
> | Top-k | Latency      | Throughput     | Accuracy     |
> |-------|--------------|----------------|--------------|
> | Top-3 | 14.08±0.56   | 547.67±13.10   | 86.43±1.21   |
> | Top-5 | 15.86±0.48   | 796.05±16.98   | 87.88±1.23   |
> | Top-7 | 17.01±0.64   | 1011.89±39.92  | 88.43±1.54   |
> | Top-10| 17.94±0.30   | 1332.07±60.05  | 88.68±1.60   |
>
> **Q3. In Table 3&4, please add the original performance.**
>
> **R3:** Thank you for your suggestion. We have added the original performance of the baselines, including Chain-of-Thought, to Table 3 and Table 4 of the revised paper.

---

### Official Review · Reviewer_bEsz · 2025-11-03

**Soundness:** 3
**Presentation:** 3
**Contribution:** 3
**Rating:** 6
**Confidence:** 4

**Summary:**

This paper proposes PPCV (Paraphrastic Probing and Consistency Verification), a two-stage framework for improving LLM reasoning by identifying and leveraging "critical tokens" - tokens that significantly influence subsequent reasoning steps. The method first uses paraphrased versions of questions to identify critical tokens where the model's predictions diverge, then substitutes these tokens with alternatives and uses consistency to select the final answer.

**Strengths:**

* Novel approaches. The method of identifying critical tokens is sound and interesting. The method does not require heavy token-level annotation to get token-level classification.
* Results are promising. Improvements are observed consistently across 5 different datasets and 2 models.

**Weaknesses:**

My main concern is that the methods do not seem generalizable.

* First, it is hard to control the quality of the paraphrased questions. Although authors design careful instructions to ensure numbers are not changed, we do not have an evaluation metric to control.

* Second, the methods can only be applied to greedy decoding; otherwise, the current method will tend to select non-top-1 tokens as the critical token.

* Third, the current methods work well for one critical token but is hard to generalize to multiple critical tokens.

**Questions:**

N/A

---

> ### Author Response · Authors · 2025-11-24
> **Response 1**
>
> **W1: First, it is hard to control the quality of the paraphrased questions. Although authors design careful instructions to ensure numbers are not changed, we do not have an evaluation metric to control.**
>
> **R1:** Thank you for highlighting this important point. We fully agree that the quality of paraphrased questions is crucial for our framework, as it directly affects both the identification of critical tokens and the stability of paraphrase-based consistency verification. To ensure high-quality paraphrasing, we adopt Automatic Prompt Engineering (APE) [1] as an evaluation to optimize the paraphrasing instruction. This allows us to systematically control the quality of paraphrases rather than relying solely on manually written prompts. The APE procedure we employ consists of four steps:
>
> 1. We create a small set of original questions paired with valid paraphrased variants. Using this exemplar set, the model generates multiple candidate instructions that could produce the paraphrased outputs from the original questions.
>
> 2. Prompt each candidate instruction to the language model to generate paraphrases for the original problems and compare the mean solve rate change before and after paraphrasing.
>
> 3. Choose the instruction that maximizes the mean solve rate change.
>
> 4. Repeat the previous steps multiple times.
>
> In our experiments, we initialize the paraphrasing instruction and iteratively refine it using APE with samples drawn from the GSM8K and MATH500 training sets. With this approach, the refined paraphrasing prompt helps us reliably produce high-quality paraphrased questions. We have incorporated the relevant discussion into the revised paper.
>
> [1] Zhou, Yongchao, et al. "Large language models are human-level prompt engineers." ICLRThe eleventh international conference on learning representations. 2022.
>
> **W2: Second, the methods can only be applied to greedy decoding; otherwise, the current method will tend to select non-top-1 tokens as the critical token.**
>
> **R2:** Thank you for raising this point. We would like to clarify that the definition of critical tokens does not depend on whether the token is the top-1 prediction under greedy decoding. Instead, critical tokens are identified based on their influence on subsequent reasoning steps. In other words, what makes a token “critical” is its role in the reasoning trajectory, not whether it ranks first among the model’s logits at generation time.
>
> Moreover, in our proposed method, the identification of critical tokens fundamentally relies on token-level influence analysis rather than the decoding strategy itself. Under non-greedy decoding (e.g., temperature sampling), a non–top-1 token can still satisfy the criteria for being a critical token if perturbing or replacing it leads to meaningful changes in downstream reasoning steps or the final answer. Therefore, the notion of critical tokens generalizes beyond greedy decoding.
>
> Our method detects such critical positions and replaces them with valid alternatives to refine the model’s reasoning trajectory. The mechanism is not restricted to the top-1 token but instead evaluates the importance of the token position in the intermediate steps. Thus, even under stochastic decoding, the method can identify influential tokens and improve subsequent steps through paraphrase-based consistency and targeted refinement.
>
> **W3: Third, the current methods work well for one critical token but is hard to generalize to multiple critical tokens.**
>
> **R3::** Thank you for this valuable point. Our method indeed identifies multiple candidate critical tokens through paraphrastic probing. While our main experiments focus on selecting the most influential critical token for paraphrase consistency, the framework itself naturally extends to the multi–critical-token setting. Specifically, paraphrastic probing yields a ranked list of token positions whose perturbations cause significant shifts in the model’s reasoning trajectory. Although we select the top candidate for the primary experiments to maintain computational efficiency, the same procedure can be applied to several critical positions simultaneously. For multiple critical tokens, we can generate alternative tokens for each identified position and apply paraphrase consistency across the new rollouts. This allows the model to refine multiple segments of its intermediate reasoning steps rather than only one. We have added relevant discussions in the revised paper.

---

### Author Response · Authors · 2025-12-02
**Summary of the rebuttal**

Dear AC,

We sincerely thank you and all reviewers for the time, effort, and constructive feedback provided for our submission. For your convenience, we summarize the key points of our rebuttal and the corresponding updates to the paper..

1. **Clarification of computational cost analysis** (Reviewers MkiR, 9jC6, s9UP). We carefully reexamined the computational cost of our method and all baselines. We also analyzed the trade-off between performance and efficiency. The updated results and discussion are now included in Figures 5 and 6 and Section 5 of the revised paper.

2. **Clarification of the generalizability and technical contribution of our method** (Reviewer bEsz0,9jC6).
We further clarified the generalizability and technical contribution of our approach and added the corresponding discussion in Section 3.3 and Appendix A.2 of the revised paper.

3. **Additional experiments and ablation studies** (Reviewers MkiR, 9jC6, s9UP).
Following the reviewers’ suggestions, we conducted additional experiments and ablation studies, including evaluations on larger reasoning models, pass@k performance, and several extended ablations. The new results are provided in Appendix A.2 and A,3.

We addressed each question thoroughly and clarified the details in the revision.

We deeply appreciate the reviewers’ thoughtful feedback and your consideration.

Best regards,

The authors of Paper 10015

---

### Meta-Review · Area_Chair_RmwC · 2025-12-29

**Summary:**

This paper proposes PPCV, a two-stage inference-time framework that uses paraphrastic probing to identify “critical tokens” and refines reasoning via consistency verification. While the idea is interesting and the reported empirical gains are notable, several fundamental issues remain unresolved, substantially limiting confidence in the method’s validity, practicality, and generality.

First, the computational cost and efficiency claims are unconvincing, and this is a core issue rather than a minor presentation flaw.
Although the authors expanded the latency analysis in the rebuttal, PPCV inherently incurs substantial additional computation, including multiple forward passes over paraphrased questions and (K \times N) truncated rollouts for alternative tokens. Under realistic assumptions—e.g., when the critical token appears near the middle of the response—the second stage alone should introduce roughly (0.5 \times K \times N) times the original CoT generation cost, independent of prefilling. The reported inference times, suggesting only a modest (≈2×) slowdown over vanilla CoT on several benchmarks, remain difficult to reconcile with this complexity, even accounting for parallelism.

Second, this computation concern cannot be resolved by additional runtime breakdowns or explanations. Since PPCV explicitly scales inference-time compute, the empirical conclusions critically depend on compute-matched comparisons. Addressing this issue would require redoing the main experiments with baselines allocated the same total inference-time budget (e.g., the same (K \times N) samples). Without such comparisons, it is unclear whether the observed gains arise from the proposed critical-token mechanism or simply from increased inference-time sampling.
Second, the paper lacks systematic compute-matched baselines.
While a paraphrased majority-voting baseline is added, the evaluation still does not consistently compare PPCV against baselines using the same total number of samples. This further weakens the attribution of performance gains to the proposed method rather than additional compute.

Third, the notion of “critical tokens” is insufficiently grounded.
The paper does not provide a clear mathematical definition of critical tokens. The heuristic based on top-1 logit mismatches under paraphrased contexts lacks theoretical justification, and it remains unclear why paraphrastic probing should reliably identify causally important tokens rather than artifacts of surface-form sensitivity.

Fourth, the applicability of the method is limited.
PPCV requires access to token-level probabilities, making it infeasible for closed-source models, which are increasingly prevalent in practice. This substantially limits the method’s generality and real-world relevance.

Finally, the semantic fidelity of paraphrased questions is not rigorously ensured.
Despite using Automatic Prompt Engineering, there is no principled guarantee or quantitative evaluation that paraphrases preserve exact semantics. Subtle meaning shifts could directly affect both critical-token identification and consistency verification, raising robustness concerns.

In summary, despite promising empirical results, the method relies on heavy and insufficiently justified inference-time computation, lacks compute-fair evaluations, rests on a weakly defined core concept, and has limited applicability and robustness. These issues outweigh the demonstrated gains, and I therefore recommend rejection.

**Reviewer Concerns:**

**Addressed (partially):**

Added experiments on larger models and pass@k

Added paraphrased majority voting baseline

Expanded computational cost breakdown and ablations

Clarified paraphrasing procedure via Automatic Prompt Engineering.

**Still Outstanding:**

Core concerns on true computational cost and implausible latency claims remain unresolved.

Lack of strict compute-matched comparisons across all baselines persists.

Semantic correctness of paraphrases is not rigorously guaranteed or evaluated.

**New concerns from AC:**

The definition and justification of “critical tokens” remain heuristic and weakly grounded.

Limited applicability to closed-source models is inherent and unresolved.

**Reviewer Scores:**

Overall, although the rebuttal addressed some presentation and completeness issues, the main concerns about computational cost, compute-matched evaluation, and conceptual grounding remain unresolved, so reviewers’ scores (6,4,6,4) would likely remain mixed.

---

### Decision · Program_Chairs · 2026-01-26

Reject